# Rewarding Doubt: A Reinforcement Learning Approach to Calibrated Confidence Expression of Large Language Models

**David Bani-Harouni**[1,2]*  **Chantal Pellegrini**[1,2]*  **Paul Stangel**[1]  **Ege Özsoy**[1,2]

**Kamilia Zaripova**[1,2]  **Nassir Navab**[1,2]  **Matthias Keicher**[1,2]

[1]Technical University of Munich
[2]Munich Center for Machine Learning
`{david.bani-harouni,chantal.pellegrini}@tum.de`

## Abstract

A safe and trustworthy use of Large Language Models (LLMs) requires an accurate expression of confidence in their answers. We propose a novel Reinforcement Learning approach that allows to directly fine-tune LLMs to express calibrated confidence estimates alongside their answers to factual questions. Our method optimizes a reward based on the logarithmic scoring rule, explicitly penalizing both over- and under-confidence. This encourages the model to align its confidence estimates with the actual predictive accuracy. The optimal policy under our reward design would result in perfectly calibrated confidence expressions. Unlike prior approaches that decouple confidence estimation from response generation, our method integrates confidence calibration seamlessly into the generative process of the LLM. Empirically, we demonstrate that models trained with our approach exhibit substantially improved calibration and generalize to unseen tasks without further fine-tuning, suggesting the emergence of general confidence awareness. Our code is available at `https://github.com/pasta99/RewardingDoubt`.

## 1 Introduction

In human intelligence and inter-human interaction, the ability to understand our own uncertainty and communicate our doubts to others is fundamental for effective decision-making, collaboration, and learning (Cosmides & Tooby, 1996; Xiong et al., 2024). Similarly, for Large Language Models (LLMs) to be safely used in real-world applications, especially when humans and AI systems work together, they must not only generate accurate information but also communicate their confidence in that information. While LLMs have demonstrated impressive capabilities in natural language understanding, question answering and text summarization (Touvron et al., 2023; Chiang et al., 2023; Achiam et al., 2023), LLMs still face significant limitations, such as their tendency to generate inaccurate information, often referred to as hallucinations (Hadi et al., 2023). This raises concerns about their reliability, particularly in real-world applications where trustworthiness is essential. Especially in high-stakes environments such as medical diagnosis, where LLMs are starting to become support tools for professionals (Moor et al., 2023; Pellegrini et al., 2025; Tu et al., 2024; Bani-Harouni et al., 2024), overconfident predictions including factual errors or hallucinations could have serious consequences for patient health. Also, in customer service or legal consultation (Shi et al., 2024; Sun et al., 2024), LLMs need to express uncertainty and defer complex queries to human representatives when unsure to avoid misinformed decisions. Reliable confidence estimation and expression would enable these systems to flag uncertain outputs for human review, ensuring that crucial decisions are not made based on uncertain LLM outputs. To allow risk estimation while using LLM-generated output, model confidence should be calibrated, meaning that the expressed numerical confidence should be equal to the probability of the model's answer being correct.

---

*Equal contribution

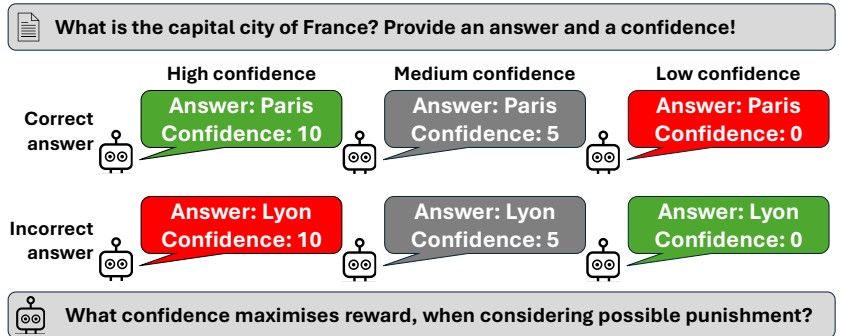

Figure 1: Illustration of our reward design: The model is rewarded for a high confidence if it is correct and punished if it is incorrect. To maximize the reward, the model needs to learn when to predict a higher or lower confidence, considering a possible higher punishment. Our reward function is designed so that the best reward is given when the confidence and the epistemic probability of being correct are the same, thus incentivizing the expression of calibrated confidences.

Many previous methods for confidence estimation lack in calibration performance as they do not train the model and instead infer the confidence from the internal state in a zero-shot setup (Huang et al., 2023; Kuhn et al., 2023; Duan et al., 2024). Additionally, this does not give models an inherent awareness of confidence. Other trained methods in this area decouple the uncertainty estimation from the text generation process (Azaria & Mitchell, 2023; Kapoor et al., 2024). This approach optimizes for calibrated confidence estimation but does not enable the uncertainty-awareness and expression in the model itself.

Targeting these limitations, we propose a novel reinforcement-learning (RL) approach for teaching LLMs to express their calibrated confidence, encouraging a granular, accurate estimation of the confidence level in the training objective. For this, we model confidence estimation as a betting game: a high-confidence answer would warrant a larger bet, reflecting a strong belief in its correctness, while a lower confidence score would suggest caution. Central to our method is a reward function based on the logarithmic scoring rule, a strictly proper scoring rule. We are the first to optimize this function through reinforcement-learning-based policy optimization, leveraging its calibration properties for directly and seamlessly training confidence calibration in LLM generations. This reward function captures the fundamental risk-reward tradeoff in probabilistic decision-making, as illustrated in Figure 1. It increases the reward when a correct answer is given with high confidence, simulating the higher potential return of big bets. Conversely, it penalizes incorrect answers more when they are made with high confidence, discouraging overconfidence. This ensures that both uncertainty and confidence are appropriately factored into the reward system. As a proper scoring rule, optimizing the reward function trains the model to align its predicted confidence with the accuracy of its output, encouraging granular and calibrated confidence scoring. A calibrated confidence estimation will provably result in the highest reward during training. This not only improves the trustworthiness of LLMs in collaborative human-AI scenarios but also helps users better assess when AI tools should be trusted, double-checked, or deferred to human expertise.

## 2 RELATED WORKS

### 2.1 CONFIDENCE ESTIMATION IN LLMS

Confidence estimation and calibration have a long history in machine learning and natural language processing (Wang, 2024). With the rise of LLMs, research has focused on adapting and extending these ideas to modern architectures. Broadly, methods fall into black-box and white-box approaches (Geng et al., 2024).

**Black-box methods** Black-box methods rely only on model outputs. Linguistic prompting methods ask the model to verbalize its confidence, sometimes aided by chain-of-thought reasoning (Xiong et al., 2024; Wei et al., 2022). Consistency-based approaches estimate confidence by mea-

suring agreement across multiple generations, with high variance indicating uncertainty (Manakul et al., 2023; Wang et al., 2022). Recently, Zhou et al. (2025) proposed SteerConf, which does multiple inference passes where the LLM is prompted to use different levels of caution in its confidence expression. The resulting verbalized confidences are aggregated based on confidence and answer consistency to an overall confidence prediction. Black-box methods are valuable for their simplicity, ease-of-use and universality, however generally lack behind white-box methods in their calibration performance.

**White-box methods** White-box methods exploit internal model states. Logit-based techniques estimate confidence from token probabilities or entropy (Huang et al., 2023; Kuhn et al., 2023; Duan et al., 2024), assuming that high probability tokens correspond to high confidence predictions. Self-evaluation methods let the model judge the truth of its own answers (Kadavath et al., 2022). They prompt the model to provide an answer followed by a judgment whether its own answer is "true" or "false". They then compare the probability of the "true" or "false" token to calculate a confidence estimation. External probing approaches train classifiers on hidden states to predict correctness (Azaria & Mitchell, 2023). While some of these methods achieve good confidence estimation results, they do not teach the model to express clear confidence values itself but depend on some auxiliary estimation mechanism.

## 2.2 FINETUNED CONFIDENCE EXPRESSION

A growing line of work integrates confidence estimation into instruction tuning. These methods typically follow a two-step paradigm: First, they estimate model confidence using various methods, e.g., self-consistency (Cheng et al., 2024; Yang et al., 2024; Han et al., 2024), token probabilities (Chen et al., 2024), trained probes (Mielke et al., 2022), empirical accuracy (Zhang et al., 2024; Lin et al., 2022; Ulmer et al., 2024), or topic unfamiliarity (Wan et al., 2024; Kang et al., 2024). Second, they construct finetuning datasets that either replace uncertain answers with refusals (Zhang et al., 2024; Cheng et al., 2024; Yang et al., 2024; Wan et al., 2024) or append the estimated uncertainty as an additional supervised signal (Han et al., 2024; Chen et al., 2024; Mielke et al., 2022; Lin et al., 2022; Ulmer et al., 2024).

The key limitation of this approach is that the model's expressed confidence is bounded by the quality of the constructed ground-truth estimates. Additionally, while the underlying confidence estimation method might optimize for perfect calibration (e.g. in the case of the trained probe), this theoretical guarantee is lost when performing supervised finetuning on these constructed ground truths to reproduce these scores.

## 2.3 REINFORCEMENT LEARNING FOR CONFIDENCE EXPRESSION

Reinforcement Learning from Human Feedback (RLHF) has proven effective for aligning LLMs with human preferences (Ouyang et al., 2022), and has also been explored for agentic interaction in textual environments (Zhou et al., 2023; Carta et al., 2023). Only recently have researchers begun applying RL directly to confidence estimation. Tao et al. (2024) adapt RLHF by designing rewards that align verbalized confidence with preference ratings, but this requires human-annotated preference data and does not address factual calibration. Leng et al. (2024) identify that standard reward models in RLHF are biased toward high verbalized confidence, rating answers with high confidence expressions with a high reward. To counteract this, they introduce two reward model training paradigms, PPO-M and PPO-C, which fine-tune the reward model to reward answers where correctness and confidence expression are aligned. Xu et al. (2024) propose RL from Knowledge Feedback (RLKF) to encourage refusals outside the model's knowledge scope, reducing hallucinations but without quantifying confidence. Stengel-Eskin et al. (2024) propose LACIE, a DPO-based approach that simulates an interaction between a speaker and a listener model, rewarding accurate and honest confidence expression by aligning it with the listener's interpretation of confidence cues rather than with fact-based numerical calibration.

In contrast to previous works, our method directly optimizes for factual calibration using a theoretically grounded, proper scoring rule as the reward signal, enabling the model to develop intrinsic uncertainty awareness without requiring external preference models, knowledge supervision, or

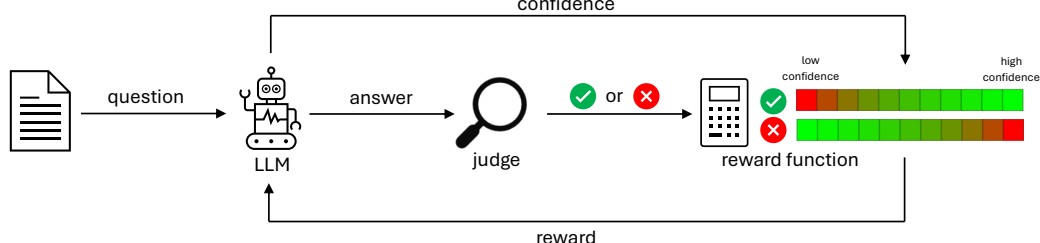

Figure 2: Overview of our reinforcement learning framework: The LLM is prompted to answer a question and provide the confidence in this answer. The answer is checked for correctness by a judge function and the reward is calculated based on the correctness and the confidence. Correct answers with high confidences are rewarded highly, but also penalized heavily when incorrect.

post-hoc calibration techniques, while at the same time seamlessly integrating calibrated confidence expression into the LLMs response generation.

## 3    REWARDING DOUBT

We propose a novel reinforcement-learning approach, that improves an LLM's ability to verbalize an accurate numerical confidence in a previously generated answer. The LLM functions as an agent in a simulated environment as shown in Figure 2, that poses challenging question-answering scenarios. It is prompted with task queries such as factual questions and asked to predict both an answer to the query as well as a confidence score. Based on the correctness of the answer, and the expressed confidence, we reward the model, incentivizing it to express a calibrated confidence.

Formally, let the model be provided with a textual question or request $q$, resulting in an answer-confidence pair $(a, \hat{p})$ as response, where $a$ is a textual answer with binary correctness value, and $0 \leq \hat{p} \leq 1$ is a numerical confidence score representing the subjective probability the model assigns to answer $a$ being correct. We train this subjective probability assessment to align with the true epistemic probability $p^*$, which represents the actual likelihood of correctness given the model's internal knowledge state. If $\hat{p}$ and $p^*$ are aligned the model is perfectly calibrated, meaning the probability of correctness $P(j(a) = 1)$ always equals the expressed confidence:

$$P(j(a) = 1 \mid \hat{p} = x) = x, \quad \forall x \in [0, 1],$$

where $j(\cdot)$ is a correctness judging function that is 1 if answer $a$ is correct, and 0 otherwise.

The true epistemic probability $p^*$ is not directly observable, thus supervised learning of calibration is only possible by constructing an artificial ground truth to approximate $p^*$. Instead, we model this task as a Markov Decision Process (MDP) defined by the tuple $(\mathcal{S}, \mathcal{A}, \mathcal{T}, R)$, where the model learns to generate calibrated confidence scores through reinforcement learning. The MDP is defined by the following components:

- **State space** ($\mathcal{S}$): A state $s_t \in \mathcal{S}$ consists of a natural language question $q$, the model's predicted answer $a$, and the partial sequence of confidence tokens predicted so far, if any. That is, $s_t = (q, a, c_{1:t-1})$, where $c_{1:t-1}$ represents the previously generated confidence score tokens.

- **Action space** ($\mathcal{A}$): The action space consists of selecting the next token $c_t$ in the confidence estimation process from the LLM vocabulary, including numerical tokens (e.g., representing percentages or probability values) and a special end-of-sequence token that finalizes the prediction.

- **Transition function** ($\mathcal{T}(s_{t+1} \mid s_t, a_t)$): The environment transitions deterministically based on the language model's autoregressive token generation process. Given a state $s_t = (q, a, c_{1:t-1})$ and an action $c_t$, the next state is defined as $s_{t+1} = (q, a, c_{1:t})$. Once the end-of-sequence token is generated, the episode terminates.

- **Reward function** ($R$): The reward $R(a, c, j)$ is computed based on the final confidence score sequence $c = (c_1, \ldots, c_T)$ and the correctness of the answer $j(a)$.

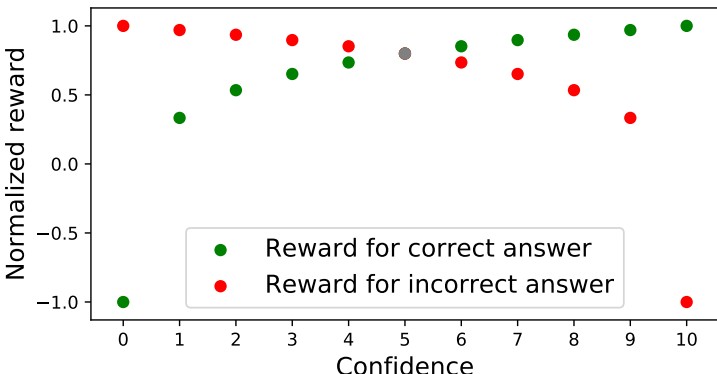

Figure 3: The rewards for each confidence value for correct and incorrect answers. The closer the confidence is to ten or zero, respectively, the higher is the reward. At the same time, the possible punishment increases to a greater extent. The model has to learn when the trade-off between those two possibilities is worthwhile.

To promote accurate confidence estimation, the model's expected reward must fulfill the requirement of being maximized when $\hat{p} = p^*$, i.e. when the predicted confidence aligns with the probability of correctness. The model should receive a high reward when it correctly predicts an outcome with high confidence or when it incorrectly predicts an outcome with low confidence. Conversely, the reward should be low when incorrect predictions are provided with high confidence or correct predictions are provided with low confidence. This approach incentivizes the model to express high confidence only in cases where certainty is warranted while expressing doubt in ambiguous situations. By penalizing both overconfidence and underconfidence, the model is encouraged to calibrate its confidence accurately, effectively balancing the trade-off between reward maximization and penalty avoidance. Note, that through this design our method focuses exclusively on improving calibration while keeping task performance stable.

We design our reward as a logarithmic scoring function:

$$R(a, \hat{p}, j) = \begin{cases} log(\hat{p}), & \text{if } j(a) = 1 \text{ (correct)} \\ log(1 - \hat{p}), & \text{if } j(a) = 0 \text{ (incorrect)} \end{cases} \tag{1}$$

This function fulfills the requirement described above as we show in the following proposition:

**Proposition 1** (Optimality implies Calibration). *The expected reward $\mathbb{E}[R(a, \hat{p}, j)]$ is maximized for each sample when $\hat{p} = p^*$ and the optimal policy under the reward design is thus perfectly calibrated.*

The proof of Proposition 1 is analogous to the proof that the logarithmic scoring rule is a proper scoring rule. We provide it in full in Appendix B and discuss the influence of the clipping on the optimality of the reward function.

Since the logarithm of zero is undefined, we introduce a small positive constant $\epsilon$ as clipping value for numerical stability. Concretely, we clip the lower and upper limit of the confidence $\hat{p}$ to $\epsilon$ and $1 - \epsilon$, respectively. The clipped reward function is provided in Appendix C. The normalized and clipped reward for correct and incorrect answers for each confidence is visualized in Figure 3.

## 4 EXPERIMENTAL SETUP

We evaluate our method in both Single-Answer and Multiple-Answer settings. We prompt the model to provide a confidence for each answer as an integer between 0 and 10, which we normalize for the reward calculation. A confidence of zero is defined as the model being certain that the answer is incorrect, while ten is defined as the model being certain the answer is correct. We normalize the reward function to the range of $[-1, 1]$.

In the Single-Answer setting we train the model on the TriviaQA dataset (Joshi et al., 2017), which contains question-answer-evidence triplets, from which we only use the questions and answers. For generalization experiments, we evaluate our method on CommonsenseQA (Talmor et al., 2019) and MedQA (Jin et al., 2020), which are multiple-choice question datasets in the commonsense and medical domain, respectively. For the Multiple-Answer setting, we train on the QAMPARI dataset (Amouyal et al., 2023), which contains questions with multiple-answers as well as evidence, again only using the questions and answers.

In the Single-Answer setting we compare our approach on the TriviaQA dataset against the following methods: Chain-of-Thought (Xiong et al., 2024), Top-K (Tian et al., 2023), Surrogate Token (Kadavath et al., 2022), Sequence Probability (Huang et al., 2023) and Self-Consistency (Wang et al., 2022) as zero-shot methods, LACIE (Stengel-Eskin et al., 2024), which uses DPO for optimizing confidence expression and Trained Probe (Azaria & Mitchell, 2023), which employs supervised training of an external probe for estimation model. We also compare to the non-finetuned base model in a zero-shot manner, using the same prompt as our Rewarding Doubt method and refer to this setup as Verbalize. In the Multiple-Answer setting we compare to Trained Probe and Sequence probability, as those methods are the best performing zero-shot and trained baselines in the Single-Answer setting. LACIE does not report results for this dataset, thus we can only compare on TriviaQA.

We report our results using the Expected Calibration Error (ECE) and the Area Under the Receiver Operating Characteristic Curve (AUROC) metric. Additionally, we visualize the calibration with calibration curves, where a well-calibrated model lies close to the 45° line and large deviations show a high miscalibration.

**Response Generation**  To calibrate and reward the model only on the confidences and not the answers we separate generation in two steps during training: Answer and confidence generation. Answers are generated first and afterwards treated as fixed inputs alongside the question, while the confidence is generated in a separate generation step and considered as sole target for optimization. Like this, we ensure that answer generation is disentangled from the optimization process, ensuring the answer correctness is not affected by our confidence calibration training.

**Correctness Assessment**  For the multiple-choice datasets MedQA and CommonsenseQA, we evaluate correctness using the exact string matching between the model's response and the ground truth answer. For the TriviaQA and QAMPARI datasets, we use the F1 score of word overlap to measure the similarity between the model's response and the ground truth candidates. The F1 score is calculated for each candidate and the maximum score is considered the final score. We consider an answer as correct if its score exceeds a threshold of 0.5.

**Implementation Details**  We optimize the reward function using the Proximal Policy Optimization (PPO) algorithm (Schulman et al., 2017). Unless stated otherwise, we use Meta-Llama-3-8B-Instruct (Grattafiori et al., 2024) as base model for our experiments. We employ the 4-bit quantized performance-optimized model version by Unsloth AI (Han et al., 2023) and apply LoRA fine-tuning (Hu et al., 2022). For the Single-Answer setting we train the model for two epochs with a learning rate of 1e-5. For the Multiple-Answer setting, due to the size of the training dataset and the fact that each question yields multiple facts, the model is trained for a limited amount of 24,000 steps with a batchsize of eight and a learning rate of 1e-5 and multiply the reward with 5 to increase its spread. All models are trained on one Nvidia A40 with each training run taking seven days. On average the model generated approximately 3.4 answers per fact. If the model fails to generate an answer in the specified format, it is penalized with an out-of-format reward of -3. Detailed implementation choices for the baselines are provided in Appendix D.

## 5  RESULTS AND DISCUSSION

This section presents and discusses the key findings of our experiments for both Single and Multiple-Answer tasks and the generalization to out-of-domain datasets.

To assess how well our approach improves calibration, we compare it against the zero-shot LLM baseline (Verbalize) and several established methods in both Single-Answer and Multiple-Answer

Table 1: Comparison of methods on the TriviaQA dataset in the Single-Answer setting with 95% CIs in brackets. * Results are from the original paper (Stengel-Eskin et al., 2024) and include standard error.

| Method | ECE ($\downarrow$) | AUROC ($\uparrow$) | Accuracy ($\uparrow$) |
|---|---|---|---|
| Verbalize | 0.3459 [0.3375,0.3543] | 0.5858 [0.5778,0.5936] | 0.6310 [0.6222,0.6397] |
| Chain-of-Thought | 0.3065 [0.2981,0.3157] | 0.6379 [0.6284,0.6475] | 0.6273 [0.6181,0.6363] |
| Top-K | 0.1611 [0.1529,0.1695] | 0.6673 [0.6580,0.6768] | 0.6023 [0.5936,0.6110] |
| Surrogate Token | 0.3686 [0.3595,0.3783] | 0.5923 [0.5818,0.6027] | 0.5933 [0.5844,0.6016] |
| Sequence Probability | 0.3156 [0.3074,0.3237] | 0.7804 [0.7725,0.7876] | 0.5955 [0.5864,0.6040] |
| Self-Consistency | 0.1134 [0.1066,0.1210] | 0.8213 [0.8129,0.8298] | 0.6224 [0.6131,0.6317] |
| PPO-M | 0.3262 [0.3173,0.3346] | 0.5274 [0.5227,0.5319] | 0.5749 [0.5662,0.5835] |
| PPO-C | 0.3607 [0.3524,0.3697] | 0.5439 [0.5384,0.5491] | 0.5258 [0.5164,0.5358] |
| LACIE* | 0.1200 $\pm$0.02 | 0.7200 $\pm$0.02 | n/a |
| Trained Probe | **0.0189** [0.0147,0.0275] | 0.8173 [0.8099,0.8250] | 0.5925 [0.5834,0.6017] |
| Rewarding Doubt (ours) | 0.0226 [0.0176,0.0302] | **0.8592** [0.8523,0.8664] | 0.6309 [0.6222,0.6399] |

question-answering tasks. Results for the Single-Answer setting on TriviaQA are presented in Table 1, and those for the Multiple-Answer setting on QAMPARI appear in Table 2. Across both tasks, Rewarding Doubt substantially improves the model's confidence calibration over zero-shot verbalization.

In the Single-Answer setting on TriviaQA, Rewarding Doubt achieves an ECE of 0.0226 and an AUROC of 0.8592, clearly outperforming all zero-shot baselines as well as LACIE, which is based on DPO-based optimization. The second fine-tuned method, Trained Probe, which relies on supervised fine-tuning, reports a slightly lower ECE (0.0189), both methods achieve near-perfect results. Further the AUROC of Rewarding Doubt is notably higher, suggesting that although both methods offer strong calibration, Rewarding Doubt better discriminates between correct and incorrect answers. In the Multiple-Answer setting on QAMPARI, Rewarding Doubt also outperforms baselines, achieving an ECE of 0.0816 and an AUROC of 0.6947. In comparison, Verbalize, Sequence Probability, and Trained Probe perform notably worse. Our findings support the claim by Azaria & Mitchell (2023) that a model's internal state encodes information about the truthfulness of statements, which can serve as an indicator of uncertainty. However, without fine-tuning, the model struggles to utilize this internal information effectively. Our approach enables the model to make use of this correlation and translate it into an accurate expression of the probability that a given answer is correct.

The calibration curves in Figure 4 further illustrate these improvements. For both TriviaQA and QAMPARI, the fine-tuned model's confidence much more closely aligns with the ideal 45° line than the zero-shot Verbalize baseline. Additionally, we observe a shift in the confidence distribution after fine-tuning. As shown in Figure 5, in a zero-shot setting the LLM (Verbalize) predominantly assigns high confidence scores (8 or above), reflecting overconfidence, a pattern also noted by Xiong et al. (2024), who attribute it to supervised pretraining that favors confident expressions. After fine-tuning with Rewarding Doubt, the model's confidence scores (shown in Figure 5b) span a wider range, including lower values, indicating a more nuanced expression of uncertainty. This shift suggests that fine-tuning mitigates overconfidence and better aligns the model's confidence with its actual performance.

Table 2: Comparison of methods on the QAMPARI dataset in the Multiple-Answer setting with 95% CIs in brackets.

| Method | ECE ($\downarrow$) | AUROC ($\uparrow$) | Accuracy ($\uparrow$) |
|---|---|---|---|
| Verbalize | 0.5319 [0.5172,0.5461] | 0.6047 [0.5837,0.6267] | 0.2550 [0.2410,0.2698] |
| Sequence probability | 0.5324 [0.5225,0.5432] | 0.5942 [0.5775,0.6110] | 0.1928 [0.1829,0.2024] |
| Trained probe | 0.1117 [0.0997,0.1262] | 0.6481 [0.6241,0.6726] | 0.2233 [0.2094,0.2384] |
| Rewarding doubt (ours) | **0.0816** [0.0723,0.0951] | **0.6947** [0.6776,0.7113] | 0.2480 [0.2348,0.2609] |

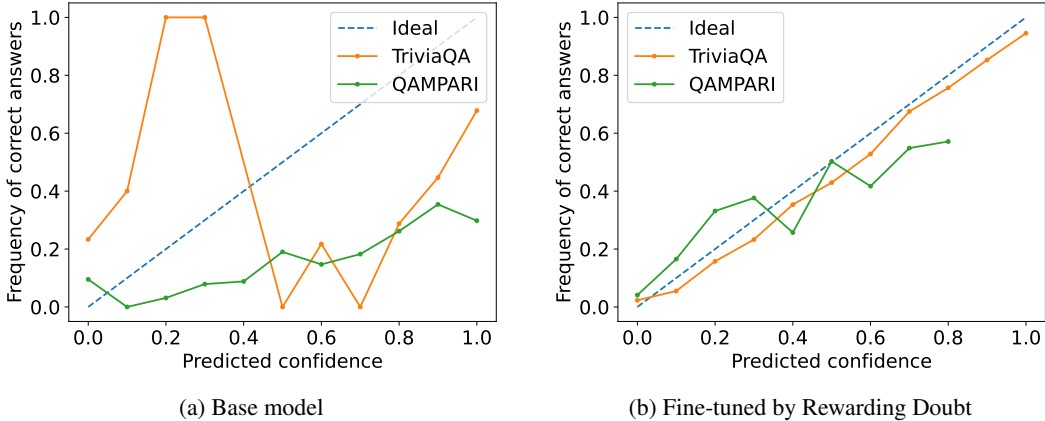

(a) Base model          (b) Fine-tuned by Rewarding Doubt

Figure 4: Calibration curves of the zero-shot base model (Verbalize) and the model fine-tuned by Rewarding Doubt.

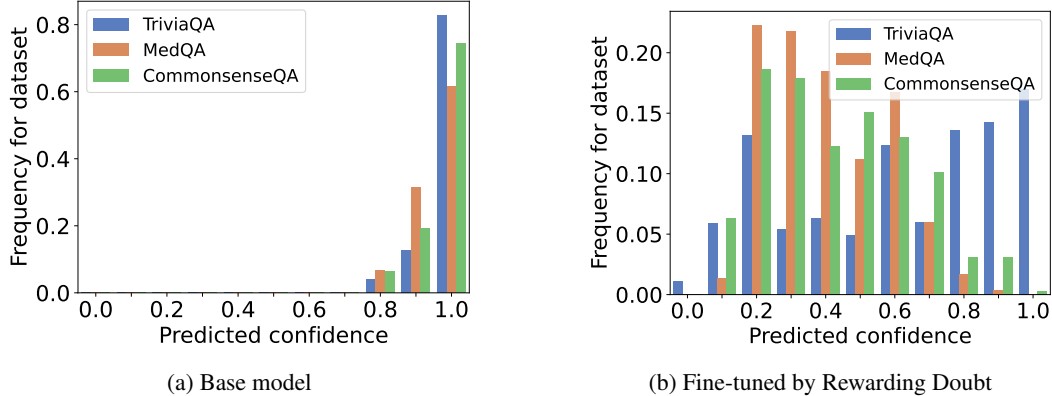

(a) Base model          (b) Fine-tuned by Rewarding Doubt

Figure 5: Histograms of predicted confidences of the zero-shot base model (Verbalize) and the model fine-tuned on the TriviaQA dataset.

To test the consistency of our method across different models, we perform an ablation study across diverse LLM architectures and sizes. Specifically, we apply Rewarding Doubt to Qwen-2.5 (3B and 7B) and Gemma-2 (9B) models, in addition to LLaMA-3.1-8B. Table 3 reports performance for each model before and after fine-tuning with our method. Despite architectural and pretraining differences, Rewarding Doubt consistently reduces calibration error and improves AUROC across all models, without degrading downstream accuracy.

**Stability of Answer Correctness**  Training confidence calibration with our method only targets the uncertainty estimation abilities and does not aim to alter the responses of the model. This is achieved by only rewarding the model on its expressed confidence, while the answer is generated beforehand independently from the model update step. Our results show a stable accuracy for all experiments without notable differences in accuracy between the base model (Verbalize) and the model adapted with Rewarding Doubt, showing that confidence calibration training with Rewarding Doubt does not affect task performance.

**Generalization Capabilities**  To assess the generalization abilities of Rewarding Doubt, we evaluated the model trained on TriviaQA in out-of-domain settings using the CommonsenseQA (Talmor et al., 2019) and MedQA (Jin et al., 2020) datasets. Results are shown in Table 4. On MedQA, Rewarding Doubt significantly outperforms Verbalize in both metrics, while on CommonsenseQA, it achieves a comparable ECE, however paired with a much higher AUROC. This discrepancy highlights a limitation of relying solely on ECE for evaluating calibration. ECE does not reflect how

Table 3: Calibration and accuracy of Verbalize vs. Rewarding Doubt across different LLMs with 95% CIs in brackets.

| Model | Method | ECE (↓) | AUROC (↑) | Accuracy (↑) |
|---|---|---|---|---|
| LLaMA-3.1-8B | Verbalize | 0.2771 [0.2689,0.2862] | 0.6766 [0.6667,0.6863] | 0.6662 [0.6577,0.6745] |
| | Trained probe | **0.0152** [0.0118,0.0235] | 0.8495 [0.8420,0.8567] | 0.6231 [0.6143,0.6322] |
| | Rew. Doubt | 0.0256 [0.0209,0.0327] | **0.8793** [0.8729,0.8860] | 0.6497 [0.6407,0.6585] |
| Qwen-2.5-3B | Verbalize | 0.5330 [0.5252,0.5435] | 0.5981 [0.5927,0.6035] | 0.4185 [0.4085,0.4247] |
| | Trained probe | **0.0186** [0.0134,0.0268] | 0.7975 [0.7880,0.8066] | 0.2540 [0.2463,0.2624] |
| | Rew. Doubt | 0.1483 [0.1415,0.1546] | **0.9065** [0.9012,0.9122] | 0.4193 [0.4097,0.4283] |
| Qwen-2.5-7B | Verbalize | 0.3619 [0.3530,0.3705] | 0.5818 [0.5762,0.5879] | 0.5239 [0.5148,0.5331] |
| | Trained probe | **0.0989** [0.0920,0.1057] | 0.8737 [0.8676,0.8797] | 0.4793 [0.4696,0.4881] |
| | Rew. Doubt | 0.1298 [0.1237,0.1368] | **0.8928** [0.8866,0.8988] | 0.5283 [0.5193,0.5368] |
| Gemma-2-9B | Verbalize | 0.3206 [0.3122,0.3288] | 0.5615 [0.5548,0.5682] | 0.6690 [0.6603,0.6773] |
| | Trained probe | **0.0301** [0.0253,0.0373] | 0.8694 [0.8629,0.8769] | 0.6464 [0.6380,0.6551] |
| | Rew. Doubt | 0.0922 [0.0861,0.0994] | **0.8649** [0.8570,0.8725] | 0.6832 [0.6743,0.6918] |

Table 4: Comparison of generalization results on CommonsenseQA (CsQA) and MedQA, trained on the TriviaQA dataset with 95% CIs in brackets.

| | Method | ECE (↓) | AUROC (↑) | Accuracy (↑) |
|---|---|---|---|---|
| CsQA | Verbalize | **0.2820** [0.2206,0.3422] | 0.5425 [0.4740,0.6069] | 0.6860 [0.6277,0.7444] |
| | Trained Probe | 0.4819 [0.4655,0.5130] | 0.5374 [0.5021,0.5708] | 0.7108 [0.6847,0.7355] |
| | Rewarding doubt (ours) | 0.2930 [0.2693,0.3179] | **0.6385** [0.6065,0.6715] | 0.7163 [0.6918,0.7410] |
| MedQA | Verbalize | 0.4480 [0.4200,0.4753] | 0.5075 [0.4803,0.5338] | 0.5067 [0.4784,0.5350] |
| | Trained Probe | 0.2099 [0.1881,0.2439] | 0.5513 [0.5207,0.5844] | 0.5051 [0.4792,0.5318] |
| | Rewarding doubt (ours) | **0.1145** [0.0893,0.1408] | **0.6649** [0.6355,0.6954] | 0.5161 [0.4886,0.5420] |

well a model discriminates between correct and incorrect predictions across different confidence levels. A model consistently assigning moderate confidence values could appear well-calibrated under ECE, yet fail to offer meaningful distinctions between uncertain and certain cases. AUROC, in contrast, directly measures this discriminative ability. Thus, the substantial improvements in AUROC underscore that Rewarding Doubt produces more useful and actionable confidence estimates. Compared to the Trained Probe, the best-performing baseline, Rewarding Doubt consistently outperforms, showing a stronger ability to generalize to new datasets.

We also explore generalization across experimental settings in Table 5 by applying a model trained in a Single-Answer setting to a Multiple-Answer task. Although under-performing a model trained specifically for that task, it still outperforms the base model considerably, demonstrating transferability of the learned confidence estimation patterns. This suggests promising applications for improving confidence estimation in more complex or less structured scenarios, such as fact verification and calibration in free-text generation, even when specialized training data is unavailable.

Our current experiments focus on settings where answer quality can be evaluated via exact rule-based metrics, yielding a binary correctness signal, the Rewarding Doubt framework could be extended to work with correctness signals provided by an LLM-as-a-judge system, a reward model trained on human preferences or continuous NLG metrics.

Overall, our experiments show that Rewarding Doubt provides a robust and efficient way to enhance calibration, while generalizing across tasks, and maintaining stable task performance, making it an effective approach for accurate confidence calibration and expression in LLMs. Beyond improvements in calibration, our method also offers practical advantages. While fine-tuning requires an initial training investment, inference remains highly efficient, as only a small, constant number of tokens need to be generated to express confidence. In contrast, zero-shot methods like Chain-of-Thought and Self-Consistency have substantial computational overhead during inference by requiring lengthy reasoning chains or multiple generations. Rewarding Doubt introduces no such over-

Table 5: Comparison of the base and fine-tuned model on the Qampari dataset in different settings with 95% CIs in brackets.

| Training | Evaluation | ECE ($\downarrow$) | AUROC ($\uparrow$) |
|---|---|---|---|
| Base model | Single fact | 0.5875 [0.5597,0.6151] | 0.5787 [0.5408,0.6125] |
| Single fact | Single fact | **0.1536** [0.1320,0.1813] | **0.7240** [0.6889,0.7577] |
| Base model | Multi fact | 0.5319 [0.5172,0.5461] | 0.6047 [0.5837,0.6267] |
| Single fact | Multi fact | 0.1777 [0.1679,0.1890] | 0.6617 [0.6468,0.6779] |
| Multi fact | Multi fact | **0.1061** [0.0935,0.1206] | **0.7268** [0.7065,0.7468] |

head, does not rely on an additional model, and directly provides actionable confidence estimates through verbalization directly by the LLM, making it highly suitable for real-world deployment.

**Limitations**   Due to computational constraints, we only tested Rewarding Doubt on models with sizes ranging from 3B to 9B parameters. While we expect similar effectiveness on larger models, empirical validation on such scales would be valuable.

## 6   CONCLUSION

In this work, we propose Rewarding Doubt, a novel approach that enables LLMs to express confidence in their answers more accurately using natural language. We leverage reinforcement learning with a reward function based on the logarithmic scoring rule that incentivizes well-calibrated confidence expressions. Fine-tuning with our method significantly improves the model's ability to estimate a calibrated confidence, effectively reducing the overconfidence patterns commonly observed in LLMs. This not only enhances the trustworthiness in AI-generated responses but also lays the groundwork for more reliable human-AI collaboration, where models can transparently communicate uncertainty, an essential step toward safer and more accountable AI systems.

## ACKNOWLEDGEMENTS

The authors gratefully acknowledge the financial support by the Bavarian Ministry of Economic Affairs, Regional Development and Energy (StMWi) under project ThoraXAI (DIK-2302-0002), and the German Research Foundation (DFG, grant 469106425 - NA 620/51-1).

## REPRODUCIBILITY STATEMENT

In order to ensure reproducibility, we describe implementation details of Rewarding Doubt as well as the used baselines in Section 4 and Appendix D. Further, Appendix A provides the exact prompts used for different experiments. Lastly, we included our code in the submission and will publish it upon acceptance.

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

# APPENDIX

# A PROMPTS

For all the question-answering settings, the model is directly prompted to answer a question without a preceding example or context. For our method the model was prompted to answer the question and additionally provide a verbalized confidence. For the other baselines that do not need a verbalized confidence but infer it indirectly, the model is prompted to only give the correct answer. The specifics for multiple-choice are slightly changed but hold mostly the same meaning. The exact prompts for each method can be seen in Table 6 for open questions and Table 7 for multiple-choice questions. The prompts for each Multi-Answer method can be seen in Table 8. We decided not to give the model a role like "expert" in the system prompt but keep it neutral, as we observed that the role we give the model affects the verbalized confidence.

Table 6: The prompts for each method for the open short form questions.

| Method | Prompt |
|---|---|
| Verbalize/ Rewarding Doubt | You will get questions. Answer with the correct answer. Additionally provide a confidence between 0, 1, 2, 3, 4, 5, 6, 7, 8, 9, 10, of how sure you are the answer is correct. A value close to 0 means you think there is a high probability that the answer is wrong. The closer the value is to 10, the higher you think is the probability that the answer is correct. The output should have the format 'Answer: <answer>, Confidence: <confidence>' and nothing else. |
| CoT | You will get questions. Read the question, analyze step by step, provide your reasoning steps, answer and your confidence in the response. The confidence should be between 0, 1, 2, 3, 4, 5, 6, 7, 8, 9, 10. A value close to 0 means you think there is a high probability that you could be wrong. The closer the value is to 5, the lower you think is the chance that you could be wrong. The output should have the format 'Thoughts: <reasoning steps>, Answer: <just the answer and nothing else>, Confidence: <confidence>'. |
| Top-K | You will get test questions with possible options. Read the question and enumerate your 5 best guesses and the probability for each that is correct (0% to 100%). Just answer with the options and nothing else. The answer should have the format 1. <answer> <probability to be correct>\n 2. <answer> <probability to be correct>\n and so on |
| Sequence Probability/ Trained Probe | You will get questions. Answer with the correct answer only and nothing else. |
| Surrogate Token | *(user)* You will get questions. Answer with the correct answer only and nothing else. *(model)* <response> *(user)* Is the proposed answer: True False \n The proposed answer is: |
| Self-Consistency | You will get questions. Read the question, analyze step by step, provide your reasoning steps and answer. The output should have the format 'Thoughts: <reasoning steps>, Answer: <just the answer and nothing else>' |

Table 7: The prompts for each method for multiple choice questions.

| Method | Prompt |
|---|---|
| Verbalize/ Rewarding Doubt | You will get test questions with possible options. Answer with the correct option. Additionally provide a confidence between 0, 1, 2, 3, 4, 5, 6, 7, 8, 9, 10, of how sure you are the answer is correct. A value close to 0 means you think there is a high probability that the answer is wrong. The closer the value is to 10, the higher you think is the probability that the answer is correct. The output should have the format 'Answer: <answer_index>, Confidence: <confidence>' and nothing else. |
| CoT | You will get test questions with possible options. Read the question, analyze step by step, provide your reasoningsteps, answer and your confidence in the response. The confidence should be between 0, 1, 2, 3, 4, 5, 6, 7, 8, 9, 10. A value close to 0 means you think there is a high probability that you could be wrong. The closer the value is to 5, the lower you think is the chance that you could be wrong. The output should have the format 'Thoughts: <reasoning steps>, Answer: <answer_index>, Confidence: <confidence>' and nothing else. |
| Sequence Probability/ Trained Probe | You will get test questions with possible options. Answer with the correct option index only and nothing else. |
| Surrogate Token | *(user)* You will get test questions with possible options. Answer with the correct option index only and nothing else. *(model)* <response> *(user)* Is the proposed answer: True False \n The proposed answer is: |
| Self-Consistency | You will get test questions with possible options. Read the question, analyze step by step, provide your reasoningsteps and the correct option index. The output should have the format 'Thoughts: <reasoning steps>, Answer: <answer_index>' and nothing else. |

Table 8: The prompts for each method for multiple fact questions.

| Method | Prompt |
|---|---|
| Verbalize/ Rewarding Doubt | Instructions: 1. You will get a question with multiple possible answers. 2. Enumerate all possible answers you know. After each individual answer state your confidence in this answer. The format should be 'Answer: <answer>, Confidence: <confidence> \n' for each individual answer. 3. The confidence should be an integer number between 0 and 10. 0 means you know for certain the answer is wrong. 10 means you know for certain the answer is correct. 4. Do not say anything else. Do not write multiple answers in one answer block. 5. When asked about dates, answer with the specific year. |
| Sequence Probability/ Trained Probe | Instructions: 1. You will get a question with multiple possible answers. 2. Enumerate all possible answers you know. Write each single answer in this format "Answer: <answer>\n" . 3. Do not say anything else. Do not write multiple answers in one answer block or any other comments. 4. When asked about dates, answer with the specific year. 5. Do not repeat answers. |

# B  PROOF

In the following, we prove Proposition 1 with the reward function

$$R(a, \hat{p}, j) = \begin{cases} log(\hat{p}), & \text{if } j(a) = 1 \text{ (correct)} \\ log(1 - \hat{p}), & \text{if } j(a) = 0 \text{ (incorrect)} \end{cases}$$

*Proof.* The proof is analogous to the proof that the logarithmic scoring function is a proper scoring function.

Let $f(\hat{p}) = \mathbb{E}[R(a, \hat{p}, j)]$ be the expected reward for all values of $\hat{p}$ and $p^*$:

$$f(\hat{p}) = p^* \log(\hat{p}) \ + \ (1 - p^*) \log(1 - \hat{p}).$$

Taking the first derivative w.r.t. $\hat{p}$:

$$f'(\hat{p}) = \frac{p^*}{\hat{p}} \ - \ \frac{1 - p^*}{1 - \hat{p}}$$

and setting

$$f'(\hat{p}) = 0 \implies p^*(1 - \hat{p}) = \hat{p}(1 - p^*) \implies \hat{p} = p^*$$

showing the only critical point in $(0, 1)$ of $f'$ is at $\hat{p} = p^*$.
The second derivative:

$$f''(\hat{p}) = -\frac{p^*}{\hat{p}^2} \ - \ \frac{1 - p^*}{(1 - \hat{p})^2}$$

is strictly negative for $\hat{p} \in (0, 1)$. Hence, $f(\hat{p})$ is concave and has its global maximum at $\hat{p} = p^*$. $\quad \square$

As the logarithm of 0 is undefined, we add a small constant $\epsilon$ in the reward function we use for training:

$$R(a, \hat{p}, j) = \begin{cases} log(\max(\hat{p}, \epsilon)), & \text{if } j(a) = 1 \text{ (correct)} \\ log(\min(1 - \hat{p}, 1 - \epsilon)), & \text{if } j(a) = 0 \text{ (incorrect)} \end{cases}$$

Through this clipping all confidence predictions between 0 and $\epsilon$, and 1 and $1 - \epsilon$, respectively, are rewarded equally. This leads to the model not being able to differentiate between confidence estimations within these ranges. We argue this effect is minor for a sufficiently small $\epsilon$ and can be disregarded in practice.

## C  CLIPPED REWARD FUNCTION

The clipped reward function as described in Section 3, is defined as follows:

$$R(a, \hat{p}, j) = \begin{cases} log(\max(\hat{p}, \epsilon)), & \text{if } j(a) = 1 \text{ (correct)} \\ log(\min(1 - \hat{p}, 1 - \epsilon)), & \text{if } j(a) = 0 \text{ (incorrect)} \end{cases} \quad (2)$$

where $\epsilon > 0$ is a small positive constant of 0.001 introduced for numerical stability to avoid evaluating the logarithm at zero.

## D  IMPLEMENTATION DETAILS OF BASELINES

For the Sequence Probability, we compute the average probability for each token in the response. In the Self-Consistency method, we let the model explore ten reasoning pathways, and the similarity of each resulting output is evaluated using the BERTScore metric Zhang et al. (2019). For the trained probe Azaria & Mitchell (2023), the original study introduced a custom dataset comprising short statements classified as either true or false. The model's activations in response to these statements were extracted from specific layers, and a multilayer perceptron (MLP) was subsequently trained on these activations to predict the truthfulness of the statements. To ensure a fair comparison, we adapted this methodology to better align with our data by allowing the model to generate answers to training dataset questions and then extracting its activations from the 24th layer for both the statements and their corresponding answers. The labels for each sample were determined following the same evaluation procedure as described in our evaluation framework. For the architecture of the MLP, we employed the same design as Azaria & Mitchell (2023) and train it for four epochs with a learning rate of 1e-4 until convergence. The exact prompts used for each baseline are provided in Appendix A.

## E    SOCIETAL IMPACT

This work introduces a reinforcement learning approach that enables Large Language Models (LLMs) to express calibrated confidence in their factual answers, advancing safe and trustworthy AI deployment. The method improves reliability and uncertainty awareness in LLMs, which is particularly valuable in high-stakes settings such as medicine, law, or customer support, where overconfident errors can have serious consequences. By optimizing a proper scoring rule during training, our method provides a theoretically sound and generalizable mechanism for aligning confidence with factual correctness—supporting human-AI collaboration and informed decision-making. However, expressing numerical confidence may lead users to overly trust AI systems, especially if the model is well-calibrated statistically but still wrong in important individual cases. This risk calls for careful deployment, appropriate user interfaces that contextualize model confidence, and safeguards against overreliance on AI-generated outputs.

## F    USE OF LARGE LANGUAGE MODELS

We employed ChatGPT to enhance the clarity of the manuscript by focusing on grammar corrections, shortening overly complex sentences, and providing alternative wording suggestions. All outputs were manually reviewed before inclusion, and no new technical material, code, results, or figures were generated by the tool.

