# OpenReview forum: "Rewarding Doubt: A Reinforcement Learning Approach to Calibrated Confidence Expression of Large Language Models"
_ICLR.cc/2026/Conference — ICLR 2026 Poster_

### Official Review · Reviewer_C9QF · 2025-10-23

**Soundness:** 2
**Presentation:** 3
**Contribution:** 3
**Rating:** 4
**Confidence:** 4

**Summary:**

The paper studies LLM calibration on Q&A. Their goal is to train the LLM to generate a predicted probability $\hat{p}$ of correctness in addition to its answer to the question. To do so, they use RL on a reward function that depends both on the correctness of the answer and the estimated probability, using a standard logarithmic scoring rule. Essentially, if the LLM is wrong and reports a large $\hat{p}$, or the LLM is correct and reports a small $\hat{p}$, it is penalized. It is well-known from forecasting that the scoring rule is optimized by reporting the agent's true best estimate of the correctness probability (i.e., it is a "proper" scoring rule). The authors study several Q&A benchmarks and show the following:
1. Their method is more effective in calibrating LLMs than existing methods.
2. The AUROC of their method (whether it is effective at distinguishing correct answers from incorrect answers) outperforms existing methods. 3. Their method does not  affect the Q&A accuracy of the model (i.e., fraction of correct answers).

**Strengths:**

Calibration in LLMs is an important problem, and the authors justify it well. Related work is discussed well. I think the idea of RL post-training on a proper scoring rule to incentivize accurate confidence prediction in LLMs is very natural (arguably more natural than existing methods). The choice of datasets is appropriate, as is the choice of which existing methods the authors compare to. I appreciate that the authors tested their method on several LLMs to ensure that the results are robust across models. The presentation is generally good, although I am confused about one point, which I discuss below. I also appreciated the authors' societal impact statement.

**Weaknesses:**

I have two main concerns:

1. **Joint vs separate optimization of answer choice and confidence level**. Lines 279-284 say:

> To calibrate and reward the model only on the confidences and not the answers we separate generation in two steps during training: Answer and confidence generation. Answers are generated first and afterwards treated as fixed inputs alongside the question, while the
confidence is generated in a separate generation step and considered as sole target for optimization. Like this, we ensure that answer generation is disentangled from the optimization process, ensuring the answer correctness is not affected by our confidence calibration training.

I think this is crucial, or else the model could strategically choose an answer that it knows is incorrect and report $\hat{p} \approx 0$ to guarantee high reward. However, Table 6 says that the prompt is

> You will get questions. Answer with the correct answer. Additionally provide a confidence between 0, 1, 2, 3, 4, 5, 6, 7, 8, 9, 10, of how sure you are the answer is correct...

which makes it seem like the model jointly generates the answer and confidence level. Could the authors please clarify this?

2. **No statistical significance analysis**. While the results are fairly impressive, I did not find any statistical significance analysis in the paper. Even basic error bars would significantly improve my confidence in the results.

I also have a third more minor concern:

3. I think Proposition 1 as stated is incorrect. The statement on line 242 refers to R(a,\hat{p},j) which was just defined on line 237 using the epsilon clipping. However, what the proof actually shows is that the *non-clipped* version of the reward function is maximized at \hat{p} = p*. In fact, the clipped version is not perfectly maximized at \hat{p} = p*. The authors argue informally that the epsilon clipping is negligible in practice, but the proof of the proposition still needs to align with its statement. I think what the authors want to do is have Proposition 1 state that the non-clipped version is maximized at \hat{p} = p*, then say that they add the clipping to avoid computing log 0 and it's negligible in practice.

**Questions:**

My primary question is stated above in Weakness 1. I would also be interested to hear the authors' responses to Weaknesses 2 and 3. I would be happy to raise my score if these weaknesses are addressed.

---

> ### Author Response · Authors · 2025-11-20
> **Rebuttal by Authors**
>
> We thank the reviewer for acknowledging that “the idea of RL post-training on a proper scoring rule to incentivize accurate confidence prediction in LLMs is very natural”, as well as our “choice of datasets” and “choice of which existing methods the authors compare” as “appropriate”.  In the following we provide answers to the open questions:
>
>
> **1. Joint vs separate optimization of answer choice and confidence level**
>
> Thank you for raising this important point. Separating answers and confidence optimization is indeed crucial to avoid gaming. While our prompt includes both answer generation and confidence rating, in our RL training setup, generation is a two-stage process:
>
> 1) The model first generates only an answer given the question. This answer is then frozen and treated as part of the input.
>
> 2) In a second step, the model is conditioned on (prompt, question, answer) and only the confidence tokens are generated; PPO updates are applied exclusively to this confidence trajectory.
>
> Thus, the answer policy is never updated by the calibration reward, and cannot change strategically to increase reward.
>
> **2. Statistical significance analysis**
>
> We now added 95% confidence intervals to all quantitative results and metrics. As we had to recompute some of the results, a few numerical values changed marginally, though all qualitative conclusions and comparison outcomes remain the same.
>
> **3. Correction of Proposition 1**
>
> Thank you for catching this, this was an oversight on our side. We now adapted the paper as follows: Proposition 1 is now stated for the unclipped logarithmic scoring rule, fitting to the provided proof, while we introduce clipping as a measure for numerical stability afterwards.

---

> > ### Comment · Reviewer_C9QF · 2025-11-20
> >
> > Thanks for the response. My concerns are resolved and I've raised my score accordingly.

---

> > > ### Comment · Reviewer_C9QF · 2025-11-20
> > >
> > > Also, it might be worth making this distinction more explicit in the paper:
> > >
> > > > While our prompt includes both answer generation and confidence rating, in our RL training setup, generation is a two-stage process
> > >
> > > so that other readers don't have the same confusion as me.

---

> > > > ### Author Response · Authors · 2025-11-20
> > > >
> > > > Thank you for the quick response, and we will make this point more clear in the paper.

---

### Official Review · Reviewer_mas6 · 2025-10-31

**Soundness:** 4
**Presentation:** 4
**Contribution:** 3
**Rating:** 6
**Confidence:** 3

**Summary:**

The paper proposes an RL-based fine-tuning approach that makes an LLM generate both an answer and calibrated confidence of its answer being factually correct. The approach uses the logarithmic scoring rule as the reward, which is maximized when the reported confidence matches the true probability of being correct. Experiments show that approach yields improved calibration while not having major impacts on the model's task performance.

**Strengths:**

1. The paper studies the important and interesting problem of teaching an LLM to reliably communicate calibrated confidence in its answer.
2. The training objective is based on the logarithmic scoring rule which is theoretically grounded.
3. Experiments show strong improvement in the model's calibrated confidence, significantly reducing overconfidence. Moreover, the method does not have a big influence on the model's accuracy.

**Weaknesses:**

1. The proposed method relies on a "judge" to check the correctness of the LLM's answer. This limits its applicability to questions for which the correctness of an answer can be easily verified. The approach does not handle subjective or open-ended questions.
2. While the method teaches an LLM to report a calibrated confidence score, it is not entirely clear to me how this should be used in practice. I imagine that a user would need to read the LLM's answer, interpret the confidence score, and decide whether or not to trust the answer based on the confidence score. (For example, if the LLM reports a confidence score of 7 out of 10, how should one decide whether or not to trust the LLM's answer?) This can be inconvenient and add cognitive burden to the user.

**Questions:**

1. Do the authors have an idea on how can the proposed method be combined with standard alignment objectives in LLM fine-tuning (e.g., RLHF) which use different rewards?

---

> ### Author Response · Authors · 2025-11-20
> **Rebuttal by Authors**
>
> We thank the reviewer for acknowledging that our paper “studies the important and interesting problem of teaching an LLM to reliably communicate calibrated confidence” with a “theoretically grounded [..] training objective”, leading to a “strong improvement in the model's calibrated confidence” while the “method does not have a big influence on the model's accuracy.” In the following we provide answers to the open questions:
>
> **1. Requirement of simple correctness judgements**
>
> Our method requires a verifiable correctness or quality signal, but this is inherent to any approach that aims to train and evaluate calibrated confidence: without some notion of answer quality, calibration is not defined. However, our method is flexible in how this signal is obtained; it only assumes that answers can be mapped to a numerical score. In our experiments this is done via exact-match / F1-based judges, but for subjective or open-ended tasks the same framework can use scores from, e.g., an LLM-as-a-judge or a reward model trained on human preferences (as in RLHF) that evaluates answer quality. When such signals are continuous rather than binary, a generalized reward function could be used: $$R(a, \hat{p}, j) = j(a) * log(\max(\hat{p}, \epsilon) + (1-j(a)) * log(\min(1-\hat{p}, 1-\epsilon),$$ where $j(a)$ is an arbitrary continuous measure of generation quality between 0 and 1. We added a discussion on this in the revised manuscript.
>
> **2. Use cases of numerical confidence scores**
>
> We can imagine multiple use cases of explicit numerical confidence. Firstly, thanks to the calibration enforced by our method, these scores also become directly interpretable for users: answers with confidence 0–10 can be directly mapped to empirical correctness probabilities of roughly 0%–100%. This allows users to set intuitive thresholds, e.g., “I only trust answers that are correct with at least 70% probability”. Secondly, numerical scores could enable automatic answer filtering, deferral, or routing.
>
> **3. Combination with RLHF**
>
> The models we used in our experiments have already been trained with RLHF before our fine-tuning, demonstrating that a two-stage combination works in practice. We can also imagine combining RLHF and Rewarding Doubt in one training run, e.g. by alternating the reward between batches.

---

### Official Review · Reviewer_8U17 · 2025-10-31

**Soundness:** 2
**Presentation:** 3
**Contribution:** 2
**Rating:** 6
**Confidence:** 3

**Summary:**

This paper introduces the new approach Rewarding Doubt that uses reinforcement learning to enhance large language models’ ability to express calibrated confidence scores better aligned with the actual predictive accuracy. The paper presents a practical idea and direction, with the proposed approach demonstrating effective results through experimental validation. However, the experimental depth and generalization are limited, and the applicability of the method to broader scenarios remains to be explored.

**Strengths:**

Leveraging reinforcement learning (RL) to improve the expression of confidence scores in large language models is a compelling idea, and the reward function proposed in this work is theoretically sound and well-motivated.

**Weaknesses:**

1. Limited Experiments. Experiments are confined to Knowledge-based QA tasks (TriviaQA, CommonsenseQA, MedQA, QAMPARI); no evaluation on other domains like math reasoning (e.g., GSM8K) or open-ended generation, limiting claims of general confidence awareness.
2. Constrainted Applicability. The method requires binary correctness judgments, making it unsuitable for open-ended tasks without ground truth (e.g., summarization, dialogue); no discussion or experiments on adaptations for such scenarios.
3. Potential Negative Impact. Accuracy is nearly identical pre/post-training on TriviaQA, but no broader benchmarks (e.g., MMLU) or ablation studies assess potential degradation in reasoning or generation quality due to confidence optimization.

**Questions:**

1. The paper does not clearly articulate the necessity of RL; what specific advantages does this PPO-based method offer over supervised fine-tuning (SFT) on approximate confidence labels, and why is SFT insufficient for optimizing the proper scoring rule?
2. No experiments are provided on larger models (e.g., >30B parameters) or full-parameter fine-tuning; how does the method perform at scale, and does full fine-tuning improve calibration compared to LoRA?

---

> ### Author Response · Authors · 2025-11-20
> **Rebuttal by Authors**
>
> We thank the reviewer for the constructive feedback and for recognizing our use of “reinforcement learning [...][as] a compelling idea” and the proposed reward function as “theoretically sound and well-motivated”. In the following we provide answers to the open questions:
>
> **1. Evaluation on reasoning tasks**
>
> In general, the formulation of Rewarding Doubt is independent of the task, as long as a numerical correctness measure can be derived. As proposed by Reviewer 8U17, we performed initial experiments on GSM8K [1]. Our results show that the ECE score is reduced from 0.1895 for the verbalized baseline to 0.0903 after training with Rewarding Doubt, while maintaining a stable AUROC (0.6182 to 0.6018), showing Rewarding Doubt effectively improves calibration. Importantly, we observe that GSM8K is relatively easy for our base model, which answers the majority of questions correctly and assigns very high confidences (8–10) to most predictions. In this high-accuracy regime, AUROC, which only reflects how well confidence separates correct from incorrect answers, has limited room to improve. Rewarding Doubt primarily reduces overconfidence within the high-confidence region, leading to a strong ECE reduction but only minor changes in AUROC.
>
> [1] Cobbe, Karl, et al. "Training verifiers to solve math word problems." arXiv preprint arXiv:2110.14168 (2021).
>
>
> **2. Applicability beyond binary correctness**
>
> While our experiments currently focus on binary correctness, our approach is easily adaptable to continuous measures of correctness making Rewarding Doubt applicable to most tasks. The only prerequisite is to define a numerical correctness measure, which could also be obtained by e.g. an LLM-as-a-judge system or NLG metrics such as Bleu-scores. In that case generalized reward could be defined as $$R(a, \hat{p}, j) = j(a) * log(\max(\hat{p}, \epsilon) + (1-j(a)) * log(\min(1-\hat{p}, 1-\epsilon),$$ where $j(a)$ is an arbitrary continuous measure of generation quality between 0 and 1, making it applicable to non-binary tasks as well. We added a discussion on this in the revised manuscript.
>
> **3. Potential negative impact on accuracy**
>
> We want to point the reviewer to the results provided in Table 4, where we evaluate our model trained on TriviaQA on two other datasets with other domains and task types. For both out-of-domain datasets accuracy remains stable, showing experimentally that our training does not harm the general capabilities of the LLM. This behaviour is expected as answer generation is separated from the RL update and never optimized directly, ensuring that confidence calibration does not modify task performance.
>
> **4. Why RL instead of SFT?**
>
> The true probability of an answer being correct is not observable, and therefore no groundtruth labels for confidence predictions exist. Using SFT on approximate pseudo-labels is possible, however such methods are upper-bounded by the performance of the pseudo-label creation. As the pseudo labels are only approximations of the true epistemic probability, this approach does not preserve the theoretical calibration guarantees.
>
> **5. Scaling to larger models**
>
> While scaling to larger models would be interesting, training full-parameter RL or experimenting with models >13B parameters is difficult under our computational constraints, and we acknowledge this limitation. However, Table 3 shows consistent improvements across four diverse architectures (3B–9B), indicating robustness across model families and sizes.

---

### Official Review · Reviewer_rSe4 · 2025-10-31

**Soundness:** 4
**Presentation:** 2
**Contribution:** 4
**Rating:** 8
**Confidence:** 4

**Summary:**

This work studies the problem of calibrated confidence expression in Large Language Models (LLMs). The authors propose a novel Reinforcement Learning (RL) approach to directly fine-tune LLMs to express calibrated confidence estimates alongside their answers to user questions. The experimental results suggest the emergence of a general confidence awareness ability in the models.

**Strengths:**

1. The verbalized confidence calibration of LLMs is a highly important topic, and this research holds significant practical relevance.
2. The experimental setup is comprehensive, and the analysis is in-depth.
3. The proposed RL-based method is solid. While the experiments are conducted on smaller models, the approach appears to be naturally scalable to large-scale models.
4. A key contribution is the effective transformation of a binary, difficult-to-train accuracy signal into a continuous confidence signal (distributed between 0 and 1) suitable for RL training. This represents a significant advancement in the domain of confidence calibration.

**Weaknesses:**

1. The paper lacks a comparison with some key baseline models in the verbalized confidence literature, such as [1].
2. The writing could be improved. For instance, Section 3 does not fully elaborate on the deeper implications or the interpretable motivation behind the design of the reward distribution function (shown in Figure 3). A more detailed analysis of the formula on line 237, specifically the concavity/convexity introduced by the *log* function, would be valuable. Explaining why this design encourages continuous confidence scores (e.g., 0.1-0.9) rather than binary (0 or 1) outputs would significantly enhance the paper's clarity.
3. While understandable, given the computational cost, providing results on larger models would make the findings more persuasive. (This is noted as a limitation, but it does not diminish the paper's core contributions, and my evaluation score will not be reduced on this basis.)

[1] Zhou Z, Jin T, Shi J, et al. SteerConf: Steering LLMs for Confidence Elicitation[J]. NeurIPS 2025.

**Questions:**

1. Could the formula on line 237 be numbered? This would adhere better to academic formatting conventions.

---

> ### Author Response · Authors · 2025-11-20
> **Rebuttal by Authors**
>
> We sincerely thank the reviewer for their encouraging feedback. We greatly appreciate their recognition that this work addresses “a highly important topic, and this research holds significant practical relevance,” and that our “experimental setup is comprehensive, and the analysis is in-depth.” We are also thankful that the reviewer found our “proposed RL-based method is solid” and that the “approach appears to be naturally scalable to large-scale models.” In particular, we value their observation that the effective transformation of “a binary, difficult-to-train accuracy signal into a continuous confidence signal […] represents a significant advancement in the domain of confidence calibration.” Below, we will address all your remaining concerns in detail. In the following we provide answers to the open questions:
>
> **1. Inclusion of additional baseline**
>
> We thank the reviewer for suggesting this interesting work, which we were not familiar with. As it was only recently accepted and will be presented at NeurIPS 2025, we consider it concurrent work. We included a discussion of this work in our related work section.
>
> **2. Additional details on the reward design**
>
> Our reward function is defined analogously to the logarithmic scoring rule, a common function used in forecasting, e.g, in cross entropy loss. As such it is concave over the whole input range. As the logarithmic scoring rule is a proper scoring rule, the expected values of our function is maximized when the predicted confidence is equal to the true epistemic confidence of the model, i.e., the true probability of correctness. This results in fine-grained calibration, as a policy that predominantly predicts extreme confidences would lead to a suboptimal expected reward. We clarify this in the updated manuscript.
>
> **3. Additional results on larger model**
>
> We share the reviewers interest in seeing results of our method on larger models, unfortunately we do not have the compute resources to run such experiments.
>
> **4. Numbering of reward formula**
>
> We thank the reviewer for the suggestion and added numbering to the formula.

---

### Official Review · Reviewer_aEuF · 2025-10-31

**Soundness:** 2
**Presentation:** 2
**Contribution:** 2
**Rating:** 4
**Confidence:** 3

**Summary:**

This paper proposes **Rewarding Doubt**, a reinforcement learning (RL)–based approach to encourage large language models (LLMs) to explicitly output calibrated confidence scores alongside their answers. The reward function is derived from the **logarithmic scoring rule**, penalizing overconfidence when the answer is wrong and rewarding confidence when correct. The method is evaluated on several question-answering benchmarks (TriviaQA, QAMPARI, CommonsenseQA, MedQA), showing improved **Expected Calibration Error (ECE)** and **AUROC** compared to baselines such as self-consistency, trained probes, and DPO-based calibration (LACIE). The authors argue that this approach yields more trustworthy confidence estimation and generalizes across domains.

**Strengths:**

* **Empirical improvement**: The proposed method achieves lower ECE and higher AUROC than baseline methods across several datasets.
* **Safety motivation**: The work aligns with the broader goal of improving AI trustworthiness and uncertainty awareness.
* **Generalization potential**: The cross-domain generalization results are encouraging.

**Weaknesses:**

Major Concern:
* **Limited evaluation scope**: Experiments are confined to factual and multiple-choice QA datasets, lacking tests in open-ended, multi-hop, or reasoning-intensive scenarios where uncertainty expression is more critical. The cross-domain results are encouraging, but do not convincingly demonstrate scalability beyond simple QA benchmarks.
* **Questionable necessity of RL**: It remains unclear why RL is essential. Similar effects might be achieved via supervised fine-tuning (SFT) using ground-truth confidence labels derived from accuracy statistics.
* **Artificial confidence expression**: The model is forced to output explicit numerical confidences, which may not generalize well to natural text generation or more integrated forms of uncertainty communication.

Minor Concern:
* **Inconsistency in figures**: Figure 3 has a labeling error in the confidence scale (0–1 vs. 0–10).
* **Lack of probing analysis**: It would strengthen the paper to analyze whether internal model representations after “Rewarding Doubt” training exhibit improved epistemic calibration, e.g., via **probing classifiers** on hidden states.

**Questions:**

1. Why is reinforcement learning necessary in this framework? Could the same reward structure be incorporated via supervised fine-tuning using ground-truth labels?
2. Have the authors examined how internal model representations (e.g., via **probing** or **representation similarity analysis**) change after training? Does the model’s epistemic awareness genuinely improve?
3. Would the method remain effective in more complex tasks such as long-form generation, reasoning chains, or dialogue systems where confidence may need to be expressed implicitly?
4. Could the authors clarify the inconsistency in Figure 3’s confidence scale (0–1 vs. 0–10)?
5. How sensitive is the method to the confidence-output format or discretization (e.g., integer vs. continuous confidence)?

---

> ### Author Response · Authors · 2025-11-20
> **Rebuttal by Authors**
>
> We thank the reviewer for highlighting that our “work aligns with the broader goal of improving AI trustworthiness” and acknowledging our “empirical improvement” and “encouraging” “cross-domain generalization results”. In the following we provide answers to the open questions:
>
> **1. Evaluation on reasoning tasks**
>
> In general, the formulation of Rewarding Doubt is independent of the task, as long as a numerical correctness measure can be derived. As proposed by Reviewer 8U17, we performed initial experiments on GSM8K [1]. Our results show that the ECE score is reduced from 0.1895 for the verbalized baseline to 0.0903 after training with Rewarding Doubt, while maintaining a stable AUROC (0.6182 to 0.6018), showing Rewarding Doubt effectively improves calibration. Importantly, we observe that GSM8K is relatively easy for our base model, which answers the majority of questions correctly and assigns very high confidences (8–10) to most predictions. In this high-accuracy regime, AUROC, which only reflects how well confidence separates correct from incorrect answers, has limited room to improve. Rewarding Doubt primarily reduces overconfidence within the high-confidence region, leading to a strong ECE reduction but only minor changes in AUROC.
>
> [1] Cobbe, Karl, et al. "Training verifiers to solve math word problems." arXiv preprint arXiv:2110.14168 (2021).
>
> **2. Unclear necessity of RL**
>
> The true probability of an answer being correct is not observable, and therefore no groundtruth labels for confidence predictions exist. Using SFT on approximate pseudo-labels is possible, however such methods are upper-bounded by the performance of the pseudo-label creation. As the pseudo labels are only approximations of the true epistemic probability, this approach does not preserve the theoretical calibration guarantees.
>
> **3. Numerical confidence expression**
>
> Current LLMs have shown advanced numerical capabilities integrated in natural text generation, even successfully solving math problems [2].
> We deliberately chose an explicit numerical confidence for objectivity, verifiability, and ease of use in downstream applications such as automatic answer filtering, deferral, or routing. Thanks to the calibration enforced by our method, these scores become interpretable: e.g., answers with confidence 0–10 can be directly mapped to empirical correctness probabilities of roughly 0%–100%. This allows users to set intuitive thresholds, e.g., “I only trust answers that are correct with at least 70% probability”.
>
> [2] Dubey, Abhimanyu, et al. "The llama 3 herd of models." arXiv e-prints (2024): arXiv-2407.
>
> **4. Inconsistency in figure 3**
>
> We thank the reviewer for spotting this mistake in our figure, we have corrected the axis in our updated manuscript.
>
> **5. Probing analysis after RL training**
>
> While we did not explore training a linear probe on the RL trained models, we believe the strong improvement in output calibration performance sufficiently shows that the model improved its internal representation of epistemic confidence as this expression also depends on the model’s internal layers.
>
> **6. Sensitivity to the output format**
>
> In early experiments, we explored a more fine-grained confidence scale of integers between 0 and 100 and observed the base model does not use the full scale of granular confidence levels, and achieves similar results to a scale of 0-10.

---

### Official Review · Reviewer_NHX7 · 2025-11-01

**Soundness:** 3
**Presentation:** 3
**Contribution:** 3
**Rating:** 6
**Confidence:** 4

**Summary:**

This paper introduces a RL-based approach that finetunes LLMs for producing well-calibrated confidence estimates, by optimizing confidence tokens with a reward using logarithmic scoring rule over both correctness and the estimated probability. This penalizes both overconfident wrong answers and under-confident correct answers. Results on single and multi-answer QA tasks demonstrate an improved confidence calibration compared to verbalized, consistency-based, probing, and another RL-based approach, without harming task accuracy.

**Strengths:**

* While there are a few existing works using RL training to improve calibration, directly optimizing the RL objective with a proper scoring rule is an original contribution.
* The results show a strong gain in calibration performance over the baselines across a wide range of QA tasks, without hurting task accuracy. The results also seem robust and generalize well to out-of-domain datasets.
* The approach provides a practical path to efficiently elicit calibrated confidence during inference, without requiring ensembling or sampling while substantially improving calibration.

**Weaknesses:**

* Besides LACIE, there are other potentially strong RL-base baselines [1,2] that would be helpful to include. For multi-answer setting with QAMPARI dataset, only the verbalized, sequence probability and trained probe baselines are compared. More consistency or RL-based baselines could be helpful. Also, temperature scaling could be applied on top of baselines.
* The main experiments and comparisons are done with Llama-3-8B-Instruct, but it would be helpful to include results on other model architectures and sizes (beyond comparing with verbalized confidence in the ablation).

---

[1]. When to Trust LLMs: Aligning Confidence with Response Quality

[2]. Taming Overconfidence in LLMs: Reward Calibration in RLHF

**Questions:**

* What if the model learns to game the reward by always answering incorrectly with 0 confidence? This might be an edge case that was not seen empirically (the accuracy does seem unaffected), but the current reward formulation does not explicitly handle this.
* Related to the last question, does it make sense to add a term or constant in the reward to further encourage accuracy? While separating the answer generation and optimizing only the confidence tokens make sense to minimize side effects on accuracy, a positive incentive might encourage correctness (e.g. if we allow updating the answer based on optimized beliefs/confidence).
* For the clipping, what epsilon value did you use for the experiments? How sensitive are the results with respect to epsilon?
* Why set the out-of-format penalty to −3? How frequent did the format failure occur?

---

> ### Author Response · Authors · 2025-11-20
> **Rebuttal by Authors**
>
> We thank the reviewer for acknowledging that “directly optimizing the RL objective with a proper scoring rule is an original contribution” and for highlighting the “strong gain in calibration performance over the baselines across a wide range of QA tasks”. We agree with the reviewer that our work’s “approach provides a practical path to efficiently elicit calibrated confidence during inference”. In the following we provide answers to the open questions:
>
> **1. Inclusion of additional baselines**
>
> We now include [2] into our related work section and provide a new comparison with the methods in Table 1 (PPO-M and PPO-C). Our method strongly outperforms these baselines in both ECE and AUROC. We also discuss [1] in our related work section, however a direct comparison is not straightforward, as no model checkpoints were published and replicating the method would require training from scratch on human preference data. Further, it does not directly address factual calibration but rather alignment of confidence expression with human preferences.
>
> **2. Inclusion of additional baselines in the Multiple-Answer Setting**
>
> We compare with the best performing baselines in Table 2. As the remaining methods are already performing much worse in the Single-Answer Setting, we did not include them in this setting.
>
> **3. Temperature Scaling**
>
> In the following, we provide the results of our main experiments with temperature scaling applied for all methods.
>
> | Method | ECE after Temperature Scaling | AUROC
> |---|---|---|
> Verbalize | 0.3234 | 0.5829
> Top-K | 0.0705 | 0.6652
> Sequence Probability| 0.2522 | 0.7781
> Surrogate Token | 0.3260 | 0.5942
> Self-Consistency | 0.0904 | 0.8180
> Chain-of-Thought | 0.2394 | 0.6369
> Trained Probe | 0.0737 | 0.8159
> Rewarding Doubt | 0.0360 | 0.8561
>
> After applying optimized temperature scaling, we observe notable improvements in ECE across several zero-shot baselines. However, they are still significantly worse in both ECE and AUROC compared to rewarding doubt. Overall, with temperature scaling for all methods, rewarding doubt shows the best results, achieving the lowest ECE (0.0360) and the highest AUROC (0.8561) among all methods.
>
> **4. Inclusion of additional baselines for experiments on other model families and sizes**
>
> We now include results for the trained probe, our best performing baseline, for all the different model families and sizes. While the trained probe often has a better ECE, Rewarding Doubt achieves a higher AUROC, showing that both methods generalize well to other model families and sizes.
>
> **5. Avoiding gaming of the reward function and possibility of rewarding correctness**
>
> We appreciate this observation and would like to clarify it further. Our training procedure does not reward answer tokens, but only updates the model’s predicted confidence. This practically prevents the model from strategically altering answers to game the reward. As there are no gradients towards the answer generation, there is no RL-driven pathway for the model to arrive in a state where it will always answer questions incorrectly.
> While it would be possible to add a reward that encourages correctness in the answer generation and allow updating the answer without much change to our methodology, it would be important to make sure the new rewards are designed such that gaming is still impossible.
>
> **6. Details on epsilon**
>
> We use an epsilon value of 0.001 for all our experiments. We added this information in the section on the clipped reward function in the manuscript.
>
> **7. Details on out-of-format penalty**
>
> We set the out-of-format (OOF) penalty to -3, as this value ensures that OOF generations are rewarded less than any other answer. After training with our method, we have a very low OOF rate of below 0.03%.

---

> > ### Comment · Reviewer_NHX7 · 2025-11-26
> >
> > Thanks for the rebuttal and the additional comparison. My concerns are resolved, and I have raised my score accordingly.

---

### Author Response · Authors · 2025-12-03
**Summary of the rebuttal process**

Dear Area Chair,

We thank the reviewers for their time and valuable feedback. Given the special situation, we want to take this opportunity to provide a short summary of the rebuttal process and highlight the changes and additions to our work.

**Reviewer NHX7:**

We are happy that we were able to resolve all concerns by reviewer NHX7. Following their review, we **added additional baselines** to our main comparison and to our comparison of different model families and sizes. We **provided temperature scaling results** of our method and baselines, and **resolved concerns regarding the possibility for the model to game the reward function**. We further **provided technical details** on our choice of the parameter epsilon and the out-of-format penalty. Following our rebuttal the reviewer acknowledged that their concerns were resolved and **raised their score from 6 to 8**.

**Reviewer aEuF:**

In the rebuttal, we addressed all concerns by reviewer aEuF. We **added an evaluation of our method on reasoning tasks**. Early results on the added dataset GSM8K showed that our method was able to improve calibration in this new setting. We further **clarified questions** on the necessity of reinforcement learning, numerical confidence expression, probing analyses after training and output format sensitivity.

**Reviewer rSe4:**

In our response to reviewer rSe4, we addressed all of their concerns. We **added the discussion of a concurrent work** to be published at NeurIPS 2025 to our related works section. We **clarified our explanation of our reward design** in the manuscript and explained our selection of baseline model sizes. We are happy that **reviewer rSe4 recommended acceptance with a score of 8**.

**Reviewer 8U17:**

During the rebuttal, we addressed all concerns and questions raised by reviewer 8U17. As suggested by the reviewer, we **added an initial evaluation on a math reasoning dataset** (GSM8K), showing that our method also improves calibration in this setting. We further **clarified how our method can be extended to scenarios with a continuous correctness signal**, pointed to experiments demonstrating that the LLM’s **accuracy on out-of-domain datasets** does not decrease, and **discussed why we use reinforcement learning** instead of supervised fine-tuning.

**Reviewer mas6:**

During the rebuttal, we addressed all open questions raised by Reviewer mas6. We **clarified how our method can be applied to open-ended or subjective tasks** by relying on more general quality signals (e.g., LLM-as-a-judge or reward models), explained **how our numerical confidence scores are intended to be interpreted** as empirical correctness probabilities and used in practice, and **outlined how our method can be combined with standard alignment objectives such as RLHF**, both in a two-stage setup and potentially within a joint training scheme.

**Reviewer C9QF:**

We are happy that we could address all concerns raised by reviewer C9QF. Their primary concern was the risk of reward gaming under joint optimization; we **clarified our two-stage generation procedure**, where the answer is generated first and then held fixed while only the confidence tokens are updated, **preventing such gaming**. Further, on request of the reviewer, we **provided confidence intervals for all our results**. Following our rebuttal the reviewer acknowledged that their concerns were resolved and **raised their score from 4 to 8**.

Overall **5 out of 6 reviewers recommended at least weak acceptance**, with **3 giving a score of 8**, showing a clearly positive post-rebuttal consensus.

We thank you again for the reviewers’ and AC’s efforts.

Best regards

The authors

---

### Meta-Review · Area_Chair_apaE · 2026-01-07

**Summary:**

This paper proposes an RL-based framework to fine-tune LLMs to express calibrated confidence in their answers to factual questions. The method uses a reward derived from the logarithmic scoring rule, penalizing both over- and under-confident predictions. By optimizing this objective, the model learns to output an explicit confidence (uncertainty) estimate alongside its answer. Experiments show improved calibration performance, with evidence that the learned confidence behavior generalizes across tasks and settings.

**Reviewer Concerns:**

Reviewer NHX7
- Some baselines are missing: Resolved. The authors added the requested additional baselines in the revised manuscript, and the proposed method consistently outperforms them.
- The experiments are limited to Llama-3-8B-Instruct: Resolved. The revision includes results on multiple model families and sizes, demonstrating that the observed calibration improvements are not specific to a single model.

Reviewer aEuF
- Limited evaluation scope, experiments are confined to factual and multiple-choice QA datasets: Partially resolved. The authors supplemented the evaluation with initial experiments on GSM8K, a reasoning benchmark.
- Questionable necessity of RL: Partially resolved. The authors argue that supervised fine-tuning is not feasible due to the absence of ground-truth confidence labels, and that SFT based on pseudo-labels would be fundamentally limited. However, this argument remains largely conceptual rather than empirically demonstrated.
- Artificial confidence expression (as numerical confidence): Partially resolved. The authors justify the use of numerical confidence scores as a principled and interpretable design choice, though questions about generalization to more natural forms of uncertainty expression remain.

Reviewer rSe4
- Missing key baseline: Partially resolved. The baseline suggested by the reviewer corresponds to a concurrent work; while no direct experimental comparison is provided, the authors added a discussion of this work in the revised manuscript.
- Results from larger models are required: Not resolved. The authors acknowledge this limitation and state that experiments on larger models were not feasible due to computational constraints.

Reviewer 8U17
- Limited experiments: Partially resolved.  The authors added initial experiments on reasoning benchmark (GSM8K).
- Constrained applicability due to reliance on binary correctness judgments: Partially resolved. The authors argue that, at least at the level of formalism, the approach can be extended to continuous measures of correctness.
- Potential negative impact on accuracy: Partially resolved. The authors note that, within the scope of the experiments reported in the paper, there is no clear evidence that the proposed method degrades task accuracy.

Reviewer mas6
- Reliance on Judge for checking correctness: Partially resolved. Same as above.
- Unclear practical use of the reported confidence score: Resolved. The authors clarified how the numerical confidence scores can be interpreted and used in practice, for example by mapping them to empirical correctness probabilities and enabling threshold-based decision making.

Reviewer C9QF
- Potential risk of reward hacking (e.g., generating incorrect answers with zero confidence): Resolved. The authors clarified that such gaming is prevented by their two-step training procedure, in which answer generation is frozen and only the confidence tokens are optimized.
- No statistical significant analysis: Resolved. The authors added 95% confidence intervals.

**Reviewer Scores:**

Most reviewers turned positive after the rebuttal, and several explicitly raised their scores as many of the key concerns were addressed through additional experiments and clarifications. Overall, I find this submission to be a solid and well-executed piece of work. While it could be further strengthened by including experiments on a more diverse set of tasks or on larger models, the current results already demonstrate clear value and technical soundness. I therefore recommend acceptance.

---

### Decision · Program_Chairs · 2026-01-26

Accept (Poster)